# Migrating Myofibroblastic Iliotibial Band-Derived Fibroblasts Represent a Promising Cell Source for Ligament Reconstruction

**DOI:** 10.3390/ijms20081972

**Published:** 2019-04-22

**Authors:** Silke Schwarz, Clemens Gögele, Benjamin Ondruschka, Niels Hammer, Benjamin Kohl, Gundula Schulze-Tanzil

**Affiliations:** 1Institute of Anatomy, Paracelsus Medical University, Nuremberg and Salzburg, Prof. Ernst Nathan Str. 1, 90419 Nuremberg, Germany; silke.schwarz@pmu.ac.at (S.S.); clemens.goegele@pmu.ac.at (C.G.); 2Department of Biosciences, Paris Lodron University Salzburg, Hellbrunnerstr. 34, 5020 Salzburg, Austria; 3Institute of Legal Medicine, University of Leipzig, Johannisallee 28, 04103 Leipzig, Germany; benjamin.ondruschka@medizin.uni-leipzig.de; 4Department of Anatomy, Otago School of Biomedical Sciences, University of Otago, 270 Great King Street, Dunedin 9016, New Zealand; niels.hammer@otago.ac.nz; 5Department for Orthopaedic, Trauma and Reconstructive Surgery, Charité-University of Medicine, Campus Benjamin Franklin, Garystrasse 5, 14195 Berlin, Germany; benjamin.kohl@charite.de

**Keywords:** ACL, ITB explant culture, Myofibroblasts, Scaffold, Spheroid

## Abstract

The iliotibial band (ITB) is a suitable scaffold for anterior cruciate ligament (ACL) reconstruction, providing a sufficient mechanical resistance to loading. Hence, ITB-derived fibroblasts attract interest for ligament tissue engineering but have so far not been characterized. This present study aimed at characterizing ITB fibroblasts before, during, and after emigration from cadaveric ITB explants to decipher the emigration behavior and to utilize their migratory capacity for seeding biomaterials. ITB and, for comparison, ACL tissues were assessed for the content of alpha smooth muscle actin (αSMA) expressing fibroblasts and degeneration. The cell survival and αSMA expression were monitored in explants used for cell isolation, monolayer, self-assembled ITB spheroids, and spheroids seeded in polyglycolic acid (PGA) scaffolds. The protein expression profile of targets typically expressed by ligamentocytes (collagen types I–III, elastin, lubricin, decorin, aggrecan, fibronectin, tenascin C, CD44, β1-integrins, vimentin, F-actin, αSMA, and vascular endothelial growth factor A [VEGFA]) was compared between ITB and ACL fibroblasts. A donor- and age-dependent differing percentage of αSMA positive cells could be detected, which was similar in ITB and ACL tissues despite the grade of degeneration being significantly higher in the ACL due to harvesting them from OA knees. ITB fibroblasts survived for several months in an explant culture, continuously forming monolayers with VEGFA and an increased αSMA expression. They shared their expression profile with ACL fibroblasts. αSMA decreased during the monolayer to spheroid/scaffold transition. Using self-assembled spheroids, the migratory capacity of reversible myofibroblastic ITB cells can be utilized for colonizing biomaterials for ACL tissue engineering and to support ligament healing.

## 1. Introduction

The iliotibial band (ITB) or *iliotibialis tract* is a lateral thickening and reinforcement of the fascia of the thigh (*fascia lata*), starting at the superior iliac spine and passing over the lateral femoral epicondyle to attach to Gerdy’s tubercle on the anterolateral side of the tibial bone [1]. The ITB consists of a dense, regularly arranged, fibrous connective tissue, in which sparse elongated fibroblasts are embedded [1]. The fibrous connective tissue of ITB mainly contains type I collagen fiber bundles, which run parallel, predominantly in the longitudinal and also in the crossing directions, with few embedded differentiated fibroblasts arranged in longitudinal columns [2]. The fibroblasts resident in the extracellular matrix (ECM) of the ITB are responsible for the ECM synthesis but, due to their small overall number, represent only a low percentage of the total volume. Hence, the fibroblasts in ligaments or tendons appear to be isolated from one another. However, a few studies could show that healthy ligament fibroblasts form a communicating network interconnected by cytoplasmic extensions with each other [3]. Type 32 and 43 connexins appear to have been identified to longitudinal and lateral columns, mediating load dependent collagen synthesis [4]. Like other bands, the ITB consists mainly of collagen [2,5], with collagen type I being a major structural constituent [6], and for the other ligaments, collagen types III, VI, V, XI, and XIV have been demonstrated as minor components [3]. In addition, the ECM of ligaments contains usually proteoglycans, e.g., aggrecan [7], elastin, and glycoproteins [3]. As found in other tendons or ligaments, the main alignment and microstructure of the ITB shows collagen bundles orientated in the longitudinal direction. Other typical ligament constituents have to be confirmed for the ITB. Along this major longitudinal axis of the dense collagenous tissue, the fiber bundles show a slight undulating appearance in the ITB based on the biomechanical requirements of the fibrous tissue [2,3,5]. This robust ITB tissue is known to be a useful and easy-to-harvest autograft for ligament, tendon, and labrum reconstruction [5,8,9,10,11] and attracts increasing interest for various tissue engineering purposes. Advantages of ITB tissue for the purpose of surgical defect reconstruction include the high density of connective tissue fibers and the ease of surgical harvest. Despite these advantages of the autograft and its broad field of application in reconstructive medicine such as anterior cruciate ligament (ACL) reconstruction, both the histological structure and the cellular component of the ITB are poorly understood [5,12]. It is known that ACLs contain myofibroblasts expressing alpha smooth muscle actin (αSMA) [13]. αSMA-positive cells have been shown in the fascia lata [12]; however, the presence of myofibroblasts in ITB tissue is mainly unknown. In addition, their role in the ligaments is controversially discussed: Myofibroblasts of the ligaments are involved in various processes such as healing and repair [13], fibrosis [14,15], as well as ACL tissue degeneration [16]. In addition, in other tendons and ligaments, a correlation between the age and the degree of crimping associated with the presence of αSMA positive cells was reported [14,15,16]. To gain more information on the myofibroblastic behaviour of ITB cells, we followed the αSMA expression in situ as well as during two-dimensional (2-D) and three-dimensional (3-D) cultures.

Explant cultures were often used for harvesting mesenchymal stem cells (MSCs) and fibroblasts from various tendons [17,18], and explants are also suitable to establish ITB fibroblast cultures [19]. In the past, explant and organ cultures of tendons of various origins, such as birds or mammals, have provided important information on the growth, metabolism, collagen formation, and response to injury [20]. However, the ITB fibroblasts are barely characterized [12]. To date, cultured explants have been used as testing systems to assess the effects of various agents and conditions [21,22,23,24,25]. Moreover, the explant culture presents a continuous source for harvesting primary fibroblasts with a high migratory potential. This migratory activity could help to guarantee a homogenous distribution of cells within a scaffold for tissue engineering applications. Primary cells cultured in monolayer systems including ligament- or tendon-derived fibroblasts undergo dedifferentiation, which is associated with a decreased expression of tendon and ligament markers [17,26]. Therefore, it is important to test the expression of differentiation markers and to monitor the capacity of the fibroblasts for ECM synthesis. A sufficient ECM production is required for tissue engineering purposes to allow for the *neo*-synthesis of ECM in scaffolds. Understanding the emigration processes of ITB fibroblasts from cultured explants provides the opportunity to assess whether these cells are able to maintain their differentiated phenotype. In addition, this migration process could be useful for seeding biomaterials with ITB fibroblasts. As differences between fibroblasts of different tissue origin are well-known, the investigation of adult human tendon or ligament fibroblasts and ITB-derived fibroblasts regarding intrinsic regeneration capacity, involving migration, replication, and collagen formation is desirable [20].

The aim of this given study was to characterize the expression profile of ITB fibroblasts in comparison to ACL-derived fibroblasts as typical ligamentocytes and to describe their migratory activity during outgrowth from the ITB tissue and scaffold colonization by cultured ITB cells.

Hence, the migratory potential of self-assembled ITB spheroids was utilized for scaffold seeding, and their myofibroblastic phenotype was monitored during the transition from 3-D to 2-D and to 3-D culture.

## 2. Results

### 2.1. Histological and Histopathological Analysis

Before the explant culture of the dissected tissue for cell isolation (Figure 1A–F), a hematoxylin and eosin (HE) staining revealed the typical order of ITB fibroblasts arranged in longitudinal columns between ECM fiber bundles within the tissue, very similar to the ACL tissue (Figure 1C,D). A significant correlation between age and cellularity was found. However, the interrelation between age and cellularity was not any longer significant when the immature ITB sample from the very young children was excluded (Figure 2A–D, Appendix A). Nearly no degenerative features could be detected in the ITB samples (Figure 2E). In contrast, ACL samples displayed diverse features of degeneration (Figure 2F). The morphological correlates of degeneration, frequently observed in ACLs, were chondroid metaplasia, cystic tissue changes, and collagen fiber deterioration.

No signs of autolysis due to the post mortal interval (<12 h) between tissue removal and the initiation of explant culture were found by the HE staining.

### 2.2. Myofibroblasts in Harvested ITB and ACL

To assess the numbers of myofibroblast-like cells in the tissue, in situ ITB and ACL tissue samples were immunolabeled for αSMA. Varying degrees of αSMA positive cells were detected in the samples (ranging from 5.1–49.2% for ITB, mean: 24.59 ± 16.82% and 3.2–59.8% for ACL tissue, mean: 21.29 ± 21.38%). Despite the degeneration score values being significantly higher in ACL compared to ITB tissues (ACL: median 4.5 vs. ITB: median 0.5), αSMA-positive cell numbers revealed no significant difference (Figure 3A,B). There was also no significant correlation between the grade of degeneration (ACLs derived from osteoarthritic knees) and the percentage of αSMA-positive cells in ACL or ITBs (Figure 2E,F). However, a significant correlation could be detected between the age of the donors and the percentage of αSMA-positive ITB cells (Figure 3C). The correlation between age and αSMA reactivity was not significant in ACL tissue cells (Figure 3D).

### 2.3. ITB Tissue and Cultured ITB Explants

The ITB tissue was characterized for typical ECM components and intracellular proteins using immunohistochemistry. The main ligament ECM components could be demonstrated such as collagen type I and III, fibronectin, decorin, and aggrecan. Lubricin and collagen type II were barely detectable (Figure 4). The ITB cells revealed a cytoplasmatic immunoreactivity for β1-integrin, CD44, vinculin, and F-actin (Figure 4).

The HE staining of cultured explants depicted the accumulation of the cells at the margin of the explants during a long-term explant culture. After a prolonged explant culture, the inhomogenous distribution and a diminished number of cells became evident within the explants compared to the native tissue samples (Figure 5A–C). While some of the examined explants were completely free of cells after finishing the explant culture, others contained the remaining cells only at the stumps and surface of the explant but not within the inner parts (Figure 5G, 109 days). Due to the prolonged culture time, a dissolution of the collagen fiber bundles in the explant occurred (Figure 5B,C,E–G). Twenty-day-old ACL explant cultures still contained cells in the core but fewer cells, which were more irregularly distributed compared with the native tissue. Most of the cells accumulated at the margins, and elongated cells obviously migrating to the margins could be detected (Figure 5B).

The microscopic analysis of an ITB explant cultured for 109 days showed dividing ITB fibroblasts at the margins and the explant’s ends (the arrows in Figure 5E and zoomed in on Figure 5F).

### 2.4. ITB Fibroblasts Expressed Typical Ligament Markers

To characterize the expression profile of ITB fibroblasts during a transition from 3-D explants to a 2-D monolayer culture, explants with emigrating cells were immunolabeled. The emigrating cells exhibited a strong immunoreactivity for vimentin. The immunolabeling of collagen type I combined with that of β1-integrin showed a collagen type I expression within the explant matrix as well as in migrating cells. These migrating cells were also positive for β1-integrin. Interestingly, emigrating cells and monolayer cultured fibroblasts revealed an intense vascular endothelial growth factor A (VEGFA) signal but showed a weaker expression of stress fibers (F-actin) compared to cells at the explant’s surface (Figure 6).

In light microscopic analyses, ITB fibroblasts exhibited a morphology very similar/comparable to ACL fibroblasts (Figure 1). In explants as well as in monolayer cultures on cover slips, ITB fibroblasts expressed the immunocytochemically detected ECM components expected in ligaments and tendons (Figure 6 and Figure 7) such as collagen type I, decorin, aggrecan, fibronectin, and lubricin. The immunolabeling of collagen type III (Figure 7A) and II (not shown) was only weak. Nevertheless, no elastin was detected in the monolayer cultured cells (not shown). Furthermore, the cell surface receptors β1-integrins and CD44 could be demonstrated. Typical mesenchymal and fibroblast markers, such as tenascin C and vimentin, have also been demonstrated on both cell types, ITB as well as ACL fibroblasts. The expression profile of the tested markers in ITB and ACL fibroblasts was very similar except for the more pronounced decorin and lower lubricin immunoreactivity in ITB compared to ACL fibroblasts (Figure 7). The cultured ITB cells expressed the same components shown in situ (Figure 4) with a weaker collagen type III expression in vitro in a 2-D culture compared to in situ conditions (Figure 7N).

### 2.5. Gene Expression Analysis

Both ACL- and ITB-derived cells expressed the tendon marker scleraxis, the fibroblast marker tenascin C, the main ECM protein type I collagen, the most abundant proteoglycan in tendon/ligaments: decorin and the glycoprotein lubricin, as shown in Appendix A. There was only a trend of a higher gene expression for collagen in the ACL and for decorin and lubricin in the ITBs, which did not reach the significance level.

### 2.6. Demonstration of Multilineage Differentiation Capacity

Compared with pure MSCs, ITB cells showed a lower differentiation capacity. They did not differentiate spontaneously (Figure 8A,B). An adipogenic differentiation was also detectable starting at days 7–14 depending on the ITB cell donor (Figure 8C,D); hence clusters of cells positive for oil red were detectable as early as on day 7. An osteogenic differentiation could be confirmed by focal Ca^2+^ deposition starting at 2–3 weeks depending on the ITB cell donor and propagated over the whole culture (Figure 8E,F). The cells survived in a pellet culture under chondrogenic and non-chondrogenic conditions and expressed the chondrogenic marker collagen type II, starting at day 7 (Figure 8G–O).

### 2.7. Cell Viability and αSMA in 2-D and 3D Cultures

Viable cells were detected homogeneously distributed within the freshly prepared explants. On the surface of the examined explants, some dead cells could be detected (Figure 9). ITB fibroblasts survived for several months in an explant culture, continuously migrating out of the explant tissue and forming novel monolayers (Figure 1, Figure 5 and Figure 6), which could be expanded for further experiments. During a prolonged explant culture of ITB tissue, fibroblasts migrated along the longitudinal collagen fiber bundles or circled laterally the bundles and multiplied at the bottom of the explants. The live/dead staining of the explants suggests a high vitality of the emigrating cells. The migrated cells formed a monolayer consisting of spread and flat fibroblast-like cells with long cytoplasmic extensions (Figure 9). By means of the hanging drop method, compact spheroids were generated by self-assembly after 72 h of incubation. The solid spheroids exhibited a high vitality of ITB fibroblasts in live/dead staining and had a diameter of 451.7 ± 26.6 µm. After a transfer to the PGA scaffolds, the ITB fibroblasts migrated out of the spheroids again and into the PGA scaffolds. After 7 days in a dynamic culture, the cells showed a transmigration through the scaffold matrix and, thus, a mostly homogeneous distribution in the biomaterial. Furthermore, the staining reflected mostly viable cells (Figure 9).

The spheroids and scaffold cultures were also immunolabeled for αSMA to assess the myofibroblastic transition. A cytoskeletal analysis revealed a pronounced αSMA expression, especially in fibroblasts just emigrating from the explants. Furthermore, these immunolabelings showed that αSMA-positive cells were located mainly at the edges of the explants. These regions appear to contain the migration active cells or to be the starting points of the migration. In contrast to the monolayer cultures, where the cells were positive for αSMA, its expression decreased in spheroid and scaffold cultures (Figure 9). Spheroids were also immunolabeled for typical ECM components and intracellular proteins. The main ligament ECM components shown in situ could also be demonstrated such as collagen types I and III, fibronectin, decorin, and aggrecan. Lubricin was barely detectable (Figure 10D). ITB cells revealed a cytoplasmatic immunoreactivity for β1-integrin, CD44, vimentin, and vinculin (Figure 10). The F-actin stress fibers were barely detectable (Figure 10K).

## 3. Discussion

Subsequent inflammatory, proliferative, and remodeling processes characterize the healing of various ligaments such as the medial collateral ligament [3]. These processes lead to the formation of an often functionally inferior scar tissue [3]. In contrast, in the case of a complete rupture and when the synovial coverage is damaged, the ACL apparently forms no bridging reparative tissue and, hence, exhibits no sufficient healing response, although initially, cellular repair processes are observed [27]. This necessitates the surgical reconstruction of the ACL mainly performed by hamstring or patellar tendon autografts [28]. In some cases, ITB tissue is used as an ACL graft [29].

Additionally, the cellular aspects of ACL reconstruction are of utmost concern besides biomechanical and structural factors [30]. The study presented here was carried out to investigate the morphology, the migratory potential, and the phenotype of ITB fibroblasts isolated by an outgrowth culture compared to ACL ligamentocytes with regard to potential biomedical applications. To date, the explant culture is a well-accepted method for the isolation of various cell types, such as MSCs from equine and human tissues (tendons/ligaments, bone marrow, as well as adipose tissue) and umbilical cord [18,20]. The explant culture is an established system not only for the isolation of tendon-derived cells but also for the study of various drug effects, such as the effects of marimistat, a generic matrix metalloproteinase (MMP) inhibitor, or actinonin, an aggrecanase inhibitor, on tendon aggrecan catabolism [31].

In this given present study, we could show that ITB fibroblasts develop a migratory activity in explant culture and thereby migrate to the tissue surface and stumps of the explants, accumulating there. Furthermore, ITB cell divisions, suggesting proliferation, were detected at these regions of the explants. We found that the majority of ITB fibroblasts leave the explant at the stumps of the ligament tissue. Additionally, we observed that the cells accumulating at the end of the explants, as well as the emigrated cells, were positive for αSMA and vimentin. Some outgrown cells floated away from the explant and adhered on the petri dish in a wider distance from the explant. The question is how the cells were activated to leave the explant, probably by hypoxia and malnutrition in the center of the ligament piece or the biomechanical stress during cutting, which maybe resembles a rupture of the tissue. A high VEGFA expression, detected in cells when leaving the explant, might support this hypothesis, since the tissue lost its vascularization after harvesting and it is known that the distance of a cell to a blood vessel or another source of nutrient supply should not be longer than around 200 µm [32].

Accordingly, it has been shown that a hypoxia-induced HIFα expression, which mediates VEGFA upregulation, is involved in the myofibroblast transition of peridontal ligament fibroblasts induced by TGFβ1 [33]. TGFβ1 is known to stimulate myofibroblast transition in tendon and ligament fibroblasts [34,35] probably via a type I receptor activin receptor-like kinase (ALK) 1 activation [36,37]. However, no growth factors were added to the culture medium in this study. Myofibroblast formation, detectable by the expression of αSMA, can contribute to pathological tissue fibrosis [37] and could be detected in degenerated ACLs [38]. Nevertheless, myofibroblasts displaying an αSMA expression could also be detected in native ITB as a part of the deep fascia of the thigh (*fascia lata*) [39,40] and ACL tissue [13]. In contrast to the study of Hasegawa et al. [38], the expression of αSMA in ACLs from OA and ACL reconstruction patients did not correlate with a histopathological degeneration in our study. Hence, the question of whether myofibroblasts contribute to ligament degeneration still remains a matter of debate. The percentage of αSMA positive cells also did not significantly differ between ACL and ITB tissue despite the degree of tissue degeneration being significantly higher in the ACL compared with the ITB donor cohorts, which represented a mostly unaltered tissue source even in older donors.

It is known that, in tendons and ligaments, αSMA expressing myofibroblasts are normally responsible for crimp formation [13,14] and, apparently, are expressed in an age-dependent manner [38]. In the two children-derived samples investigated in the present study, we found also a high number of αSMA-expressing cells and more crimps. If these samples were removed, the correlation was not any longer significant. The results of the present study could also confirm the age dependency of an αSMA immunoreactivity in ITB tissue with higher numbers of αSMA-positive cells in younger individuals in ITB tissue. This significant correlation could not be demonstrated in ACL tissue samples, probably due to the smaller age range and higher mean age of the donor collective available, not including immature samples. ITB tissue reflected a higher cellularity in samples from very young children compared to older donors. Interestingly, the observation of a lower cellularity in samples from older compared to younger donors as reported by others [41,42] could only be confirmed here for the case that the very young donor was included into the analyses. Myofibroblasts have been implicated in tendon/ligament healing [27,30,43], mediating the contraction phase of healing where healing ligaments achieve their original length [44]. Hence, Laumonier and colleagues induced a myofibroblast transition by treating ITB fibroblasts with TGFβ1 to transplant these activated cells in ligaments to support their healing process [43]. In our study, the ligamentocytes were activated during emigration from explants, probably requiring contractile forces and showing an increased αSMA expression in the subsequent monolayer culture with nearly all cells positive for αSMA. However, their αSMA immunoreactivity decreased after introducing them in 3-D cultures such as spheroids and PGA scaffold cultures, underlining their dynamic plasticity and suggesting a reversibility of the αSMA expression. The downregulation of the αSMA expression might be associated with a decreasing activity. Interestingly, it has been reported that fibroblasts derived from tenopathic tendons maintained an altered expression signature (in regard to podoplanin, CD106, and CD248), suggesting an activation which persisted in vitro and led also to a reaction profile which differed from healthy fibroblasts in response to IL-1β treatment [45]. This indicates the advantage of having the opportunity of harvesting ligamentocytes from unaffected tissue sources such as the ITB.

In response to fibroblast emigration, cell distribution becomes inhomogeneous within the cultured explant. Near the surface of the explant, more dividing cells were found than in the core, but dying cells were also histologically detected in these superficial areas. Stress fibers became clearly evident in cells leaving the explants. Fibroblasts leaving the explant lost their order (formerly within the tissue in longitudinal rows), exhibited a flattened shape, and spread larger with long cell extensions. In a study by Lee and Ehrlich [22], it has been shown that, by a vanadate treatment of chicken embryonic tendon explants, the tendon fibroblasts were stimulated to maintain their typical order during emigration from an explant. The authors hypothesized that the myosin light chain phosphorylation supported the deposition of aligned collagen fibers by the fibroblasts [22]. In emigrating cells, collagen was irregularly deposed. Using an explant culture could lead to the harvest of a distinct migratory cell population, whereas other subpopulations might not leave the explant. It is known that tendons contain stem cell populations with differing properties [46]. Hence, the question arose whether cells with a stem cell character showing a multilineage potential might emigrate. We found osteogenic, adipogenic, and chondrogenic differentiation potentials of isolated ITB cells. Tendon-derived stem cells were usually isolated by enzymatic methods [46].

Moreover, the condition of the explant culture of the present study without any mechanical loading should be taken in account, since the unloading influences the gene expression profile in tendon explants [4,23]. It has been shown that the expression of ECM remodelling enzymes changes due to an unloading in the explant culture [23]. Some changes in the non-collagenous protein content proteoglycans but not of collagen have been detected during the longterm culture of tendon explants [47].

In this study, typical ECM components of tendons and ligaments could be demonstrated in native ITB which were still expressed in explant-derived ligamentocytes and also in 3-D spheroids. There were no major differences in the expression profiles of 2-D cultured ITB- and ACL-derived fibroblasts, neither at the protein nor at the gene expression levels. Since the ITB is used for ACL reconstruction [5], this observation is of outstanding importance regarding the structural functionality of ITB autografts applied for ACL reconstruction. In addition to collagen type I, the major component of the tendon and ligament ECM which provides tensile strength, proteoglycans, is also present in the ECM which are of functional importance [31], and the gene expression of the glycoprotein lubricin could be shown. Samiric et al. found that about 80% of the total proteoglycans in tendon was decorin [48] and confirmed also the presence of aggrecan. In the present work, the expression of both proteoglycans, decorin and aggrecan, was confirmed at the protein level for both ITB and ACL fibroblasts. The expression of collagen type III was higher in 3-D spheroids than in the 2-D cultures of ITB cells, suggesting more similarity between spheroids and in situ conditions. In contrast to ITB cells, human ACL fibroblasts expressed fewer decorin in accordance with a previous study comparing human ACL tissue with other rabbit-derived tendons/ligaments [49]. In regard to the gene expression, a similar trend could be observed, which did not reach the significance level. However, the exact function of aggrecan present within tensional regions of tendons [7,31,50], which are exposed solely to longitudinal tension stress loads, still needs to be fully elucidated [31]. The small leucine-rich proteoglycan decorin appears to prevail in the proximal/tensional regions of tendons and ligaments and is known to participate in the regulation of collagen fibrillogenesis as well as the binding of these proteoglycans to collagen [31,48,50]. Vinculin acts as a mechano-coupling protein fixing integrin-mediated cell-matrix adhesions and provides a link between the external environment and the inner actomyosin cytoskeleton. As an essential part of focal adhesions, it is fundamental for the transmission of contractile forces for cell morphogenesis [51,52] and is expressed by ITB as well as ACL-derived fibroblasts. The slightly more pronounced expression of lubricin in ACL-derived ligamentocytes could only be shown on the protein but not at the gene expression level and could be explained by the unique intraarticular milieu of the ACL tissue.

Furthermore, with the present work, it could be shown that fibroblasts can grow out of the ITB explants over several months and can continuously be used to obtain new monolayer cultures. In explant and monolayer cultures, they expressed fibroblast typical markers such as VEGFA, β1-integrins, vinculin, CD44, and vimentin comparable to ACL-derived fibroblasts. The expression and synthesis of αSMA as well as F-actin stress fibers reveals the transdifferentiation of emigrating ITB fibroblasts to myofibroblasts.

## 4. Materials and Methods

### 4.1. Tissues

Human ITB samples were harvested less than 12 h postmortem from 13 patients (mean age: 42.2 years, between 5 weeks and 65 years; 7 male and 6 female donors included) and used for cell isolation (Figure 1A). ITB samples were derived from the distal part of the upper third of the thigh obtained during an autopsy at University of Leipzig, Institute of Legal Medicine (ethical approval number 156-10-1207-2010). ACL samples (donor: 9 females, mean age 50.11, 29–74 years), used as controls (Figure 1B), were harvested during a knee joint reconstruction or an ACL reconstructive surgery in accordance with the institutional ethical committee of the Charité-Universitätsmedizin Berlin, Campus Benjamin Franklin (ethical approval number EA4-033-08, approval date: 22 May 2008). ITB and ACL samples were primarily characterized by hematoxylin and eosin (HE) staining (Figure 1C,D). An ACL explant derived from an adult female rabbit (New Zealand rabbit, knees obtained from the abattoir) was also included, exclusively into a histological analysis (HE). Since both are bradytrophic tissues with low autolysis rates, a period of less than 12 h (mean 6.8 h) was kept between the sampling and initiation of the explant culture to exclude signs of autolysis.

### 4.2. Histology

Before and after several weeks of explant culture, the native ligament samples (ITB, length approx. 2 mm) and ITB explant samples (length: approx. 2 mm) were fixed in a 4% paraformaldehyde solution (PFA, Santa Cruz Biotechnology Inc., Dallas, USA), dehydrated, and finally embedded in paraffin. After slicing, the 7-µm sections were rehydrated in a descending ethanol series and prepared for HE as well as Alcian blue (AB) staining.

For HE staining, the sections were incubated for 6 min in Harry`s hematoxylin (Sigma-Aldrich, Munich, Germany), before being rinsed in tap water and counterstained for 4 min in eosin (Carl Roth GmbH, Karlsruhe, Germany). The sections were covered with Entellan (Merck-Millipore, Darmstadt, Germany).

For the AB performance, the sections were rehydrated in a descending ethanol series and subsequently incubated for 3 min in 1% acetic acid before being incubated for 30 min in a 1% AB staining solution (Carl Roth GmbH). After rinsing in 3% acetic acid and a 2 min washing step in distilled water (A. dest.), fibroblast cell nuclei were counterstained for 5 min in a nuclear fast red aluminium sulphate solution (Carl Roth GmbH).

### 4.3. Histopathology

Stained sections of ligaments were graded by histopathology based on HE and AB (ITB: *n* = 9 (4 female, 5 male, mean age: 35.23 years, age range: 5 weeks–64 years), ACL: *n* = 8, all female (osteoarthritis (OA) patients and healthy donors) mean age: 56.34 years, age range: 29–74 years) using the degeneration score applied and published by Hasegawa et al. in 2013 [38]. The score summarizes aspects such as collagen fiber orientation, mucoid degeneration, chondroid metaplasia, cystic changes, and inflammation (15 points maximal: 0: none, 0.5: minimal, 1: mild, 2: moderate, and 3: severe). The histopathology of pseudonymized samples was performed by an experienced anatomist, who was blinded for age, diagnosis of the donors, and the results of other analyses.

### 4.4. Fibroblast Isolation by Explant Culture and Cell Culturing

The explant cultures were prepared from ITB and ACL tissues. Surrounding the connective tissue, in the case of ACL, the synovial membranes were also removed. The pure ligament tissue was cut into 2–3 mm^2^ slices and incubated in T-25 culture flasks with a growth medium (Dulbecco’s modified Eagle’s medium (DMEM)/Ham’s F-12, 1:1) containing 10% fetal calf serum (FCS), 1% penicillin-streptomycin, 2.5 µg/mL amphotericin B, 1% nonessential amino acids (all from Biochrom AG, Berlin, Germany), and 25 µg/mL ascorbic acid (Sigma-Aldrich) at 37 °C and 5% CO_2_. After 1–2 weeks, the ITB and ACL fibroblasts started to migrate from the tissue slices (Graphical abstract). Subsequently, fibroblasts were harvested using 0.05% trypsin/0.02% EDTA (Biochrom AG) and expanded in T-75 and T-175 culture flasks (CellPlus, Sarstedt AG, Nümbrecht, Germany) (Figure 1E,F) for further characterization, spheroid, and scaffold cultures. The explants were cultured for 8–10 weeks. For analyzing the marker protein expression profile by immunocytochemistry, ITB as well as ACL fibroblasts derived from three different donors each were seeded on poly-l-lysin-coated cover slides at 1 × 10^4^ cells/cm^2^. For the gene expression analysis, three human ITB and three human ACL samples obtained from different donors were cultured in a monolayer (passages 3–4).

### 4.5. Culture of ITB and ACL Fibroblasts on Cover Slips

For analyzing the marker expression profile on protein level, cover slips were washed twice with phosphate buffered saline (PBS, Biochrom AG). After rinsing with sterile A. dest., the cover slips were placed in sterile petri dishes and covered with a poly-l-lysin solution (1:100 in 1x PBS; both Biochrom AG) for 20 min at room temperature (RT). Subsequently, the coated cover slips were rinsed with sterile A. dest. and dried at RT for 4 h. Until further use, they were stored at 4 °C.

For the vitality assay and immunofluorescence analysis, ITB (passages 1–3) as well as ACL fibroblasts (passages 1–5) were seeded on the coated cover slips with an initial density of 1 × 10^4^ cells/cm^2^ and allowed to adhere for another 24 h. Before the respective staining procedures, the medium was removed and the cells were washed in 1× PBS. For the vitality analysis, the cover slips were immediately transferred to live/dead assay, while for performing the immunofluorescence analysis, the cells were fixed in 4% PFA for 15 min and stored at 4 °C.

### 4.6. Vitality Assay

The cell vitality in the tendon explants, tenocytes cultured on cover slips, spheroids, and scaffold cultures (at least three independent experiments with each culture type and cells derived from three different donors were performed) was visualized using a live/dead assay based on fluorescein diacetate (FDA, Sigma-Aldrich) and propidium iodide (PI, Carl Roth GmbH).

The samples were incubated for 30 s in FDA/PI staining solution (9 µg/mL FDA and 10 µg/mL PI dissolved in PBS). The green (living cells, FDA) or red (dead cells, PI) fluorescence was monitored using a SPEII confocal laser scanning microscope (Leica, Wetzlar, Germany).

### 4.7. Immunofluorescence Analysis of Marker Expression

The protein expression profile was assessed using confocal laser scanning microscopy. ACL-derived fibroblasts seeded on cover slips were used as a control and compared to adherend ITB fibroblasts. Monolayers (passages 1–3 of ITB and 1–5 of ACLs) were stained for the whole marker panel. Explants only for a reduced panel of markers such as collagen type I and β1-integrin, vimentin, VEGF-A, F-actin, spheroids, and tissue paraffin sections were also stained for the marker panel (monolayer: *n* = 3, tissue: *n* = 6, spheroids: *n* = 3). The scaffolds were only stained for αSMA. Explants, spheroids (undifferentiated and chondrogenically differentiated), scaffold segments, deparaffinized sections (7-µm thick), or seeded cover slips, fixed in 4% PFA, were washed with Tris buffered saline (TBS: 0.05 M Tris, 0.015 M NaCl, pH 7.6), before being incubated with a protease-free donkey serum (5% diluted in TBS with 0.1 % Triton X 100 for cell permeabilization) for 20 min at RT. Subsequently, the samples were incubated with primary antibodies (see Table 1: collagen types I–III, aggrecan, decorin, fibronectin, elastin, tenascin C, lubricin, hyaluronan receptor CD44, β1-integrin, vimentin, αSMA, vinculin, and VEGFA overnight at 4 °C in a humid chamber. As controls, the primary antibodies were omitted or replaced with the corresponding isotype antibody (mouse IgG1), which were used in the same concentrations as the original primary antibodies. The samples were rinsed with TBS before incubation with donkey-anti-goat or anti-rabbit-Alexa-488 (Invitrogen, Carlsbad, USA) or donkey-anti-mouse- or anti-goat-cyanine-3- (Cy3, Invitrogen) coupled secondary antibodies (diluted 1:200 in TBS, see Table 1), respectively, for 1 h at RT. The cell nuclei were counterstained using 10 µg/mL 4′,6′-diamidino-2-phenylindol (DAPI, Roche, Mannheim, Germany) and phalloidin-Alexa-488 (1:100, Santa Cruz Biotechnologies Inc.) to depict the F-actin cytoskeletal actin architecture. The immunolabeled cells were washed several times with TBS before mounting with a fluoromount mounting medium (Southern Biotech, Biozol Diagnostica, Eching, Germany) and examined by using a SPEII confocal laser scanning microscope. The αSMA positive and negative cells were counted in three representative microscopic fields of tissue samples of 9 (ITB) and 8 (ACL) donors to determine the percentage of positive cells.

### 4.8. Gene Expression Analysis of Marker Expression Using Realtime Detection PCR

RNA was isolated using the RNeasy Mini Kit (Qiagen AG, Hilden, Germany) according to the manufacturer’s protocol. The quantity and purity (e.g., 260/280 absorbance ratio) of the RNA samples were monitored using the Nanodrop ND-1000 spectrophotometer (Peqlab Biotechnologie GmbH, Erlangen, Germany).

A real-time PCR analysis was carried out to compare the gene expression of typical ligament components in cultured ITB- and ACL-derived fibroblasts (each *n* = 3, passages 3–4), which are associated with the normal ligamentous phenotype. For a cDNA synthesis, 1000 ng of the total RNA was reverse transcribed using the QuantiTect Reverse Transcription Kit (Qiagen AG) according to the supplier manual. Twenty ng of cDNA was used for each quantitative real-time PCR (qRT-PCR) reaction using TaqMan Gene Expression Assays (Life Technologies) with primer pairs for scleraxis, tenascin C, type I collagen, decorin, lubricin, and HPRT as a reference gene (Table 2). qRT-PCR was performed using the real time PCR detector StepOnePlus (Applied Bioscience (ABI), Foster city, USA) thermocycler with the program StepOnePlus software 2.3 (ABI). For all used primers, the mean normalized expression (MNE) ratios, using HPRT as the reference gene, were calculated as mentioned before [53,54].

### 4.9. Multipotency Testing of ITB Fibroblasts

To test the presence of progenitor cells in the isolated ITB fibroblast populations, differentiation experiments using chondrogenic, adipogenic, and osteogenic media were performed for at least 4 weeks (*n* = 3). The ACL cells were not tested for multipotency since their differentiation capacity has been thoroughly described [55,56,57,58,59,60].

For adipogenesis, fibroblasts of passages 2–6 were seeded at 65,000 cells per cm^2^ on poly-l-lysin-coated cover slips and incubated with adipogenic media (DMEM, 50 µM dexamethasone (Sigma-Aldrich), 10 µg/mL insulin (Sigma-Aldrich), 100 µM indomethacin, 500 µM 3-isobutyl-1-methylxanthine (both: Sigma-Aldrich), 5 µM rosiglitazone (Cayman chemical company, Ann Arbot, Mitchigan, USA), 25 mM HEPES (Merck), 10% FCS, and 1% penicillin-streptomycin).

For osteogenesis, fibroblasts of passages 2–6 were seeded at 20,000 cells per cm^2^ on poly-l-lysin-coated cover slips and incubated with adipogenic media (DMEM, 1 µM dexamethasone, 10 mM glycerol-2-phosphate (Sigma-Aldrich), 200 µM ascorbic acid, 25 mM HEPES, 10% FCS, and 1% penicillin-streptomycin). Non-induced cells (10,000 cells per cm^2^) cultured in an expansion medium served as the control.

For chondrogenesis, pellet cultures consisting of 200,000 cells per pellet were either cultured with chondrogenic media supplemented with 10 ng/mL TGFβ1 (Pepro Tech GmbH, Hamburg, Germany), DMEM, 2 mM L-glutamine, 25 mM HEPES, 1 mM sodium-pyruvate (Merck), 0.1 µM dexamethasone, 0.2 mM ascorbic acid, 0.35 mM prolin, ITS+1 (Liquid media supplement, Sigma-Aldrich), or 1% penicillin-streptomycin or maintained under non-chondrogenic conditions (control) for 4 weeks. Finally, chondrogenesis was checked by immunolabeling cartilage-specific type II collagen using a specific antibody, a fluorophore-coupled secondary antibody (Table 1), and confocal laser scanning microscopy. The cell survival was tested after 4 weeks using a live/death assay.

#### 4.9.1. Alizarin Red Staining

Osteogenesis was analyzed using alizarin red staining, since alizarin red forms an orange-red deposit with calcium. The cells cultured on poly-l-lysin-coated cover slips were rinsed 2 × 5 min with PBS at RT, before being fixed for 60 min using 60% isopropanol. After rinsing in TBS, the cover slips were stained with the alizarin red solution (Sigma-Aldrich) for 5 min. The sections were dehydrated in acetone, followed by an acetone-xylene (1:1) solution, and cleared in xylene (Roth) before being covered with Entellan. The light microscopical images were taken using a Leica DM1000 LED.

#### 4.9.2. Oil Red Staining

Adipogenesis was analyzed using oil red staining. The cells cultured on poly-l-lysin-coated cover slips were rinsed for 5 min with PBS and for 5 min with A. dest. at RT, before being fixed for 60 min using 4% PFA. Then, the cells were rinsed again for 2 × 5 min using PBS and for 5 min in distilled water. The fixed cells were immersed in 60% isopropanol (5 min) and covered with the oil red staining solution (stock solution 3 g/L: diluted 3:2) for 5 min at RT. The cover slips were rinsed with tap water until clear before and after being counterstained with hematoxylin for 30 s and kept wet with tap water. Photos were immediately taken using a light microscope (DM1000 LED with ICC 50 HD camera (Leica, Wetzlar, Germany).

### 4.10. Spheroid Culture and Scaffold Seeding Using ITB Fibroblasts

A spheroid formation was achieved using the hanging drop method. A cell suspension (25,000 cells per 50 µL) was dropped on the lid of a non-coated petri dish. The lid was turned back on the bottom of the dish and cultured for 72 h until the spheroids were ready to be harvested. For scaffold seeding, 50 spheroids were placed on 1 cm^2^ poly-glycolic acid (PGA) scaffolds (PGA biofelt, 65 mg/cc, Concordia Medical, Warwick, NY, USA). The spheroids were allowed to adhere to the scaffolds for 48 h in a static culture before being transferred to a bioreactor tube (TubeSpin^®^ Bioreactor 50, TPP, Trasadingen, Switzerland) on an orbital shaker (Sunlab, Sustainable Lab Instruments, Aschaffenburg, Germany) at 36 rpm and dynamically cultivated for 7 d. Images of the stained and immunolabeled samples were taken using a SPE-II confocal laser scanning microscope.

### 4.11. Statistical Analysis

The data was expressed as mean values with standard deviations. The statistics were compiled using Graphpad Prism 6 (version 6.02, GraphPad Software, San Diego, CA, USA). A statistical significance was set at a *p* value of ≤0.05. The Rout test was used to identify outliers. A normal distribution was proved using the D’Agostino and Pearson omnibus normality test (α = 0.05). The unpaired two-tailed *t*-test parametric analysis was used to compare myofibroblast numbers. The unpaired two-tailed Mann–Whitney Test (α = 0.05) was applied for comparing the degeneration score values. The Spearman nonparametric correlation coefficient and linear regression were determined.

## 5. Conclusions

ITB fibroblasts reflect transient myofibroblastic transition and were able to self-assemble to spheroids and to emigrate from them into the PGA scaffolds. Hence, the migratory potential of ITB fibroblasts can be used for scaffold seeding. PGA which is known to be cytocompatible for tendon/ligament-derived cells [61,62] was used as a valid model system. The use of ITB fibroblast spheroids enables the maintenance and utilization of the ITB fibroblast migration capacity for the re-cellularization of biomaterials for tissue engineering, in particular, for ACL reconstruction applications.

## Figures and Tables

**Figure 1 ijms-20-01972-f001:**
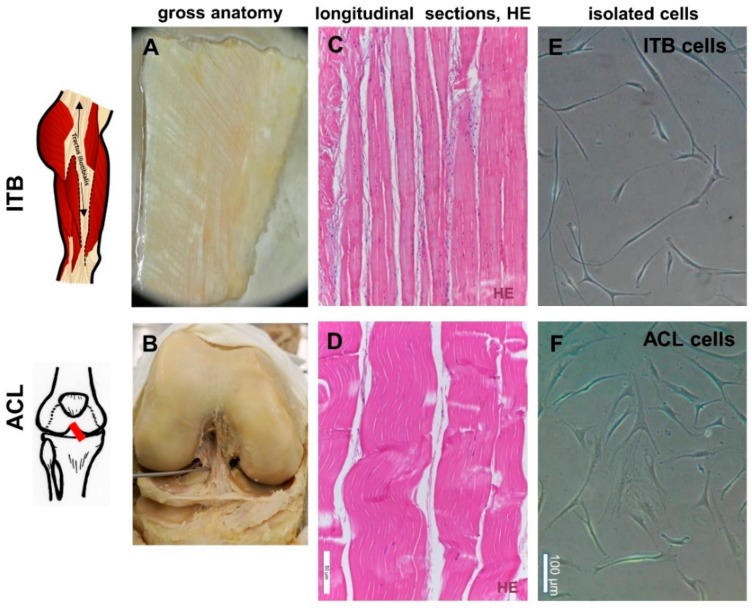
The gross anatomy, histology, and isolated fibroblasts of iliotibial band (ITB) and anterior cruciate ligament (ACL) tissue: The gross anatomy of ITB (**A**) and ACL (**B**) tissue. The hematoxylin and eosin (HE) staining of ITB (**C**) and ACL (**D**) tissue. Fibroblasts isolated from ITB (**E**) and ACL (**F**) tissues. Scale bars: 50 µm (Figure 1C,D) and 100 µm (Figure 1E,F). ITB: iliotibial band, ACL: anterior cruciate ligament.

**Figure 2 ijms-20-01972-f002:**
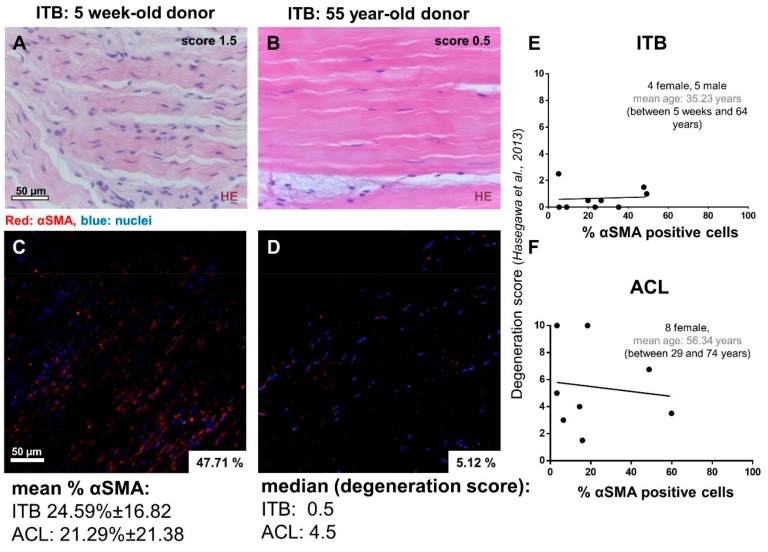
Representative ITB samples stained with hematoxylin eosin (HE) and immunolabeled for α-smooth muscle actin (αSMA). (**A**,**B**) A representative HE staining of ITB tissue of a young and aged donor. (**C**,**D**) αSMA-staining of the same samples. No significant correlation could be shown between the stage of degeneration and the percentage of αSMA expressing cells for ITB (**E**) (spearman: *r* = 0.149, *p* = 0.701) and ACL (**F**) (spearman: *r* = −0.096, *p* = 0.806) tissue samples. Scale bars: 50 µm. ITB: iliotibial band, ACL: anterior cruciate ligament.

**Figure 3 ijms-20-01972-f003:**
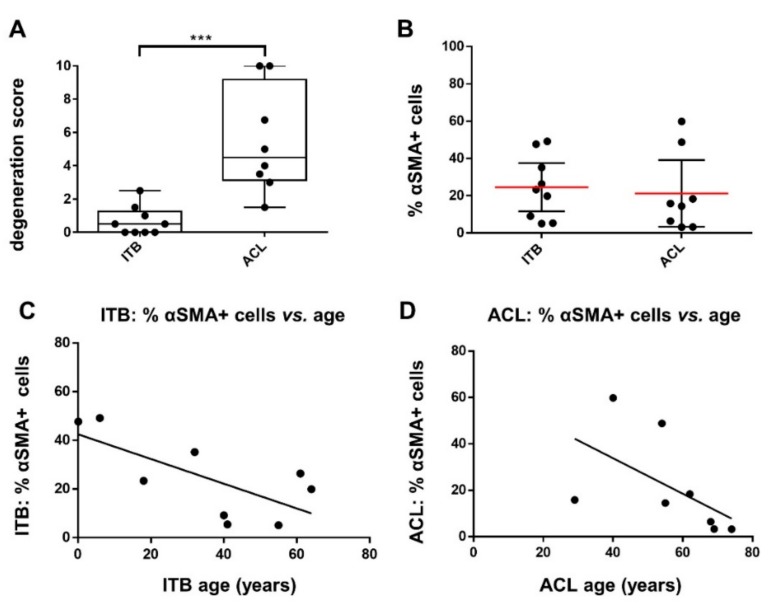
The degree of degeneration and the percentage of αSMA-positive cells as well as age vs. percentage of αSMA-positive cells in ITB and ACL. (**A**) The differences in the degeneration scoring results and (**B**) the percentage of αSMA-positive cells. The correlation between the percentage of αSMA-positive cells in ITB (**C**) and ACL tissue samples (**D**) and donor age. Figure 1A: *** *p* = 0.0002, Mann–Whitney test; Figure 1B: no significant difference; Figure 1C: significant negative correlation (*p* = 0.035), pearson, *r* = −0.703; (**D**): not significant, pearson, *r* = −0.552, *p* = 0.156.

**Figure 4 ijms-20-01972-f004:**
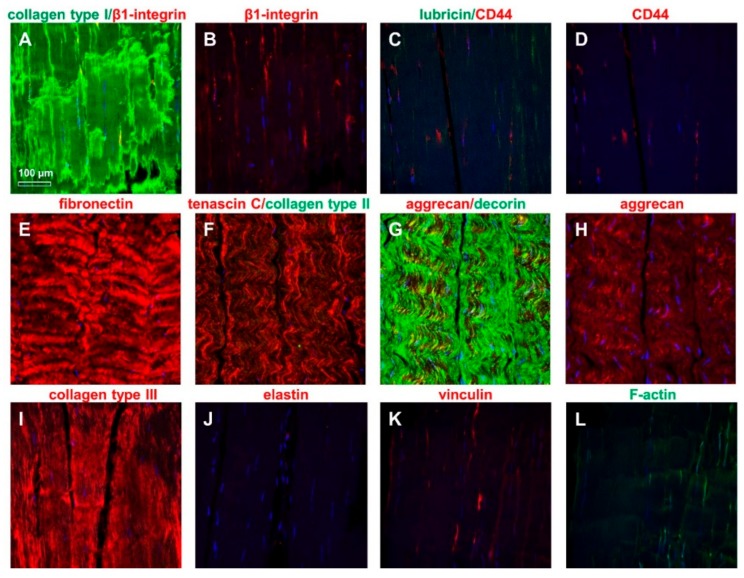
The immunolabeling of the extracellular matrix (ECM) and cytoplasmatic markers in ITB tissue: Collagen type I (green) and β1-integrin [(**A**) merged, (**B**) β1-integrin (red)], lubricin (green)/CD44 (red) [(**C**) merged, (**D**) CD44], fibronectin [(**E**) red], tenascin C (red)/collagen type II (green) (**F**), aggrecan/decorin [(**G**) merged, (**H**) aggrecan], collagen type III [(**I**) red], elastin [(**J**) red], vinculin [(**K**) red], and F-actin [(**L**) green]. The cell nuclei were counterstained using 4’,6-diamidino-2-phenylindole (DAPI). Scale bar: 100 µm.

**Figure 5 ijms-20-01972-f005:**
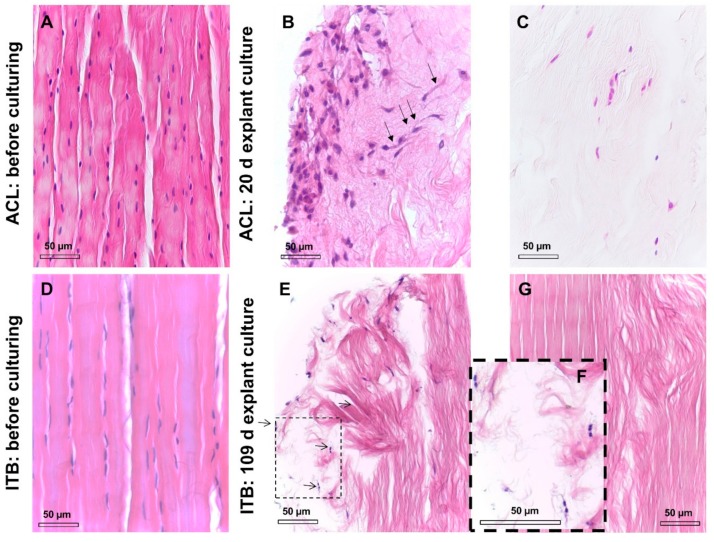
The HE staining of ACL and ITB tissue before and after prolonged explant culture. The tissue before culturing: (**A**) rabbit-derived ACL and (**D**) human ITB tissue. (**B**) The margin and (**C**) center of a rabbit-derived ACL explant after 20 days of culturing. In Figure 5B, the migration of ACL cells to the margin is visible (arrows). (**E**) The margin and (**G**) center of a human-derived ITB explant after 109 days of culturing. Cell divisions (zoomed in the inset (**F**) suggest cell proliferation (arrows). In Figure 5C, the core of the ACL tissue contained few inhomogeneously distributed cells after 20 days of culturing. In Figure 5G, the central part of the same ITB explant contained no cells after 109 days of culture. Scale bars: 50 µm. ACL: anterior cruciate ligament, ITB: iliotibial band.

**Figure 6 ijms-20-01972-f006:**
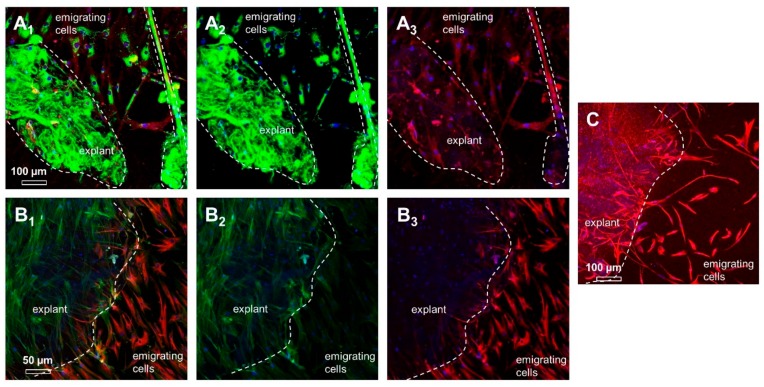
The expression profile in ITB explants and emigrating cells. (**A**) The main ligament extracellular matrix (ECM) protein collagen type I and the corresponding cell ECM adhesion receptor β1-integrin are shown (**A_1_**: merged, **A_2_**: collagen type I (green), and **A_3_**: β1-integrin (red)). Dashed line: The explant and, on the right side, a thick bundle of collagen fibers are shown. (**B**) F-actin filament organization and synthesis of vascular endothelial growth factor A (VEGFA) in explants cultured for 5 weeks and emigrating cells (**B_1_**: merged, **B_2_**: F-actin (green), and **B_3_**: VEGFA (red)). (**C**) The mesenchymal intermediate filament vimentin (red). Scale bars: Figure 6A,C: 100 µm and Figure 6B: 50 µm.

**Figure 7 ijms-20-01972-f007:**
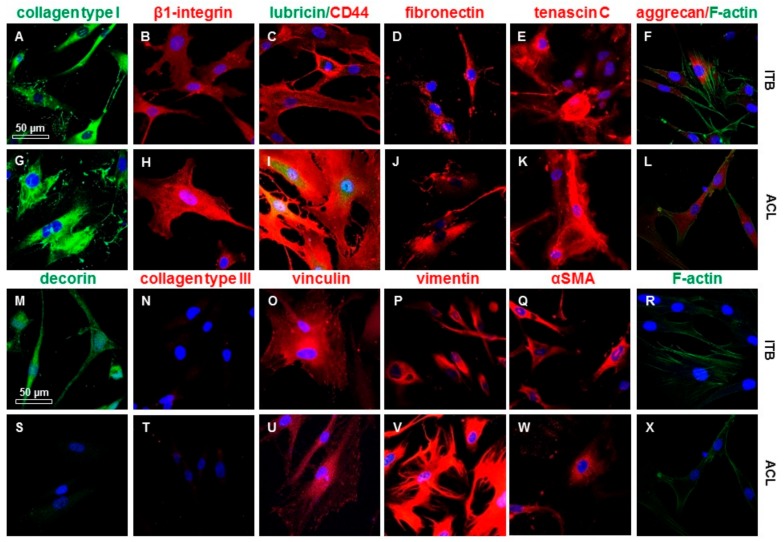
The expression profile of 2-D cultured ITB and ACL fibroblasts. The Iliotibial band (ITB): monolayer passage (P)3 (**A**–**F**, **M**–**R**) and anterior cruciate ligament (ACL): P5 (**G**–**L**, **S**–**X**). Collagen type I (**A**,**G**), β1-integrin (**B**,**H**), lubricin and CD44 (**C**,**I**), fibronectin (**D**,**J**), tenascin C (**E**,**K**), aggrecan (**F**,**L**), decorin (**M**,**S**), collagen type III (**N**,**T**), vinculin (**O**,**U**), vimentin (**P**,**V**), alpha smooth muscle actin (αSMA) (**Q**,**W**), and F-actin (**R**,**X**). Blue: cell nuclei. Scale bars: 50 µm.

**Figure 8 ijms-20-01972-f008:**
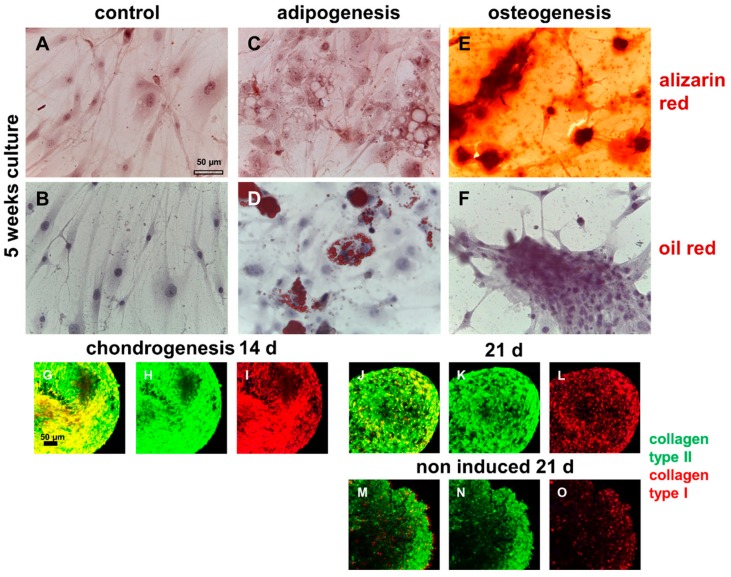
The multilineage differentiation potential of isolated migrative ITB cells. (**A**,**C**,**E**) Alizarin red staining. (**B**,**D**,**F**) Oil red staining. **A**,**B**: Non-induced control. **C**,**D**: Adipogenically induced. **E**,**F**: Osteogenically induced. Chondrogenically induced for 14 d (**G**–**I**) or 21 d (**J**–**L**) and noninduced for 21 d (**M**–**O**). **G**,**J**,**M**: Merged (collagen type I: red, collagen type II: green); **H**,**K**,**N**: collagen type II (green); **I**,**L**,**O**: collagen type I (red). Scale bars: 50 µm. ITB, iliotibial band.

**Figure 9 ijms-20-01972-f009:**
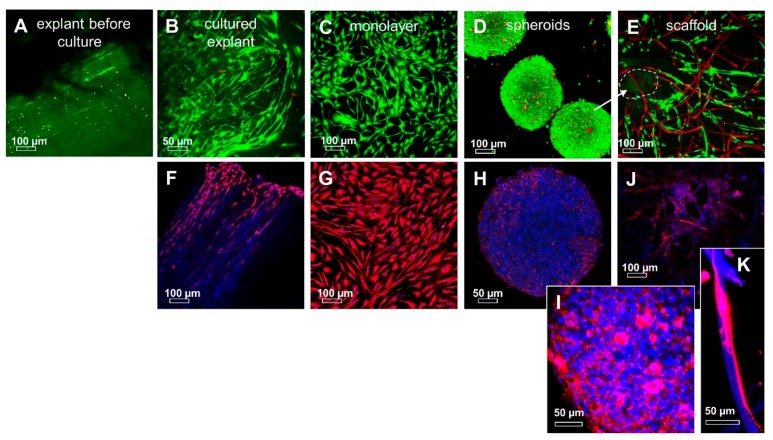
The vitality and αSMA immunoreactivity of ITB tissue and 2-D and 3-D cultured ITB fibroblasts. (**A**) Freshly prepared explant (before cultured); (**B**) explant after 109 days of culturing; (**C**) monolayer, P0; (**D**) mini spheroids consisting of 25.000 ITB cells after 3 days of spheroid formation; and (**E**) polyglycolic acid (PGA) scaffold colonized by ITB fibroblasts for 7 days. ITB cells emigrated from spheroids placed on the scaffold (**E**): the arrow and dashed line mark the position of the former spheroid) and spread on the fibers. Vital cells: green, dead cells or scaffold fibers: red. The αSMA immunoreactivity (red) is shown (**F**) in an explant after 109 days of culturing; (**G**) monolayer, P0; (**H**) mini spheroids consisting of 25.000 ITB cells after 2 days of spheroid formation ((**I**) zoomed); (**J**) PGA scaffold colonized by ITB fibroblasts for 3 days; (**K**) zoomed inset. αSMA immunoreactive cells: red. Cell nuclei: blue. Some degrading fibers stain also blue. Scale bars: 100 µm (**A**–**G**, J) and 50 µm (**H**,**I**,**K**).

**Figure 10 ijms-20-01972-f010:**
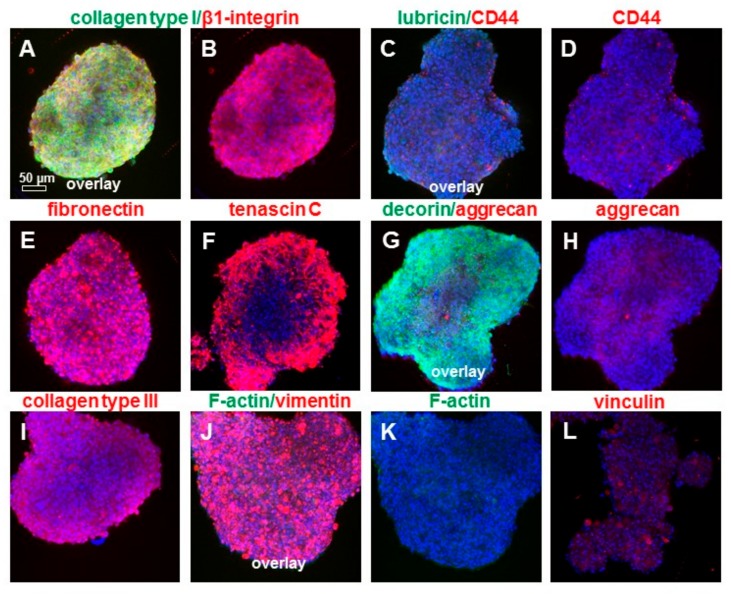
The immunolabeling of ECM and cytoplasmatic markers in ITB spheroids. Mini-spheroids consisting of 25.000 ITB cells after 3 days of spheroid formation were immunolabeled for collagen type I (green) and β1-integrin (red) (**A**,**B**), lubricin (green) and CD44 (red) (**C**,**D**), fibronectin (**E**) (red), tenascin C (**F**) (red), decorin (green) and aggrecan (red) (**G**,**H**), collagen type III (**I**) (red), and vimentin (red) as well as stained with phalloidin-488 to visualize F-actin fibers (green) (**J**,**K**); and immunolabeled for vinculin (**L**) (red). The cell nuclei were counterstained using DAPI (blue). Scale bar: 50 µm.

**Table 1 ijms-20-01972-t001:** The antibodies used.

Target	Primary Antibody	Dilution	Secondary Antibody	Dilution
Aggrecan	mouse antihuman R&D systems, Minneapolis, USA	1:30	donkey-anti-mouse cy-3, Invitrogen	1:200
CD44	mouse-antihuman, Cell signalling Technology, Danvers, USA	1:50	donkey-anti-mouse cy-3, Invitrogen	1:200
Collagen type I	goat anti human, Abcam, Cambridge, UK	1:50	donkey anti goat, Alexa Fluor 488, Invitrogen, Carlsbad, USA	1:200
Collagen type II	rabbit anti human, Acris Laboratories, Hiddenhausen, Germany	1:50	donkey anti rabbit, Alexa Fluor 488, Invitrogen	1:200
Collagen type III	mouse anti human Acris Laboratories, Hiddenhausen, Germany	1:30	donkey-anti-mouse cyanine-3 (cy3), Invitrogen	1:200
Decorin	rabbit anti human, Acris Laboratories, Hiddenhausen, Germany	1:30	donkey anti rabbit, Alexa Fluor 488, Invitrogen	1:200
Elastin	mouse anti human Acris Laboratories, Hiddenhausen, Germany	1:30	donkey-anti-mouse cy-3, Invitrogen	1:200
Fibronectin	mouse-antihuman, Dianova, Hamburg, Germany	1:30	donkey-anti-mouse cy-3, Invitrogen	1:200
β1-integrin	mouse-antihuman, Merck-Millipore	1:30	donkey-anti-mouse cy-3, Invitrogen	1:200
Lubricin	rabbit-antihuman, Abcam, Cambridge, UK	1:30	donkey anti rabbit, Alexa Fluor 488, Invitrogen	1:200
αSMA	mouse-antihuman, Sigma-Aldrich	1:50	donkey-anti-mouse cy-3, Invitrogen	1:200
Tenascin C	mouse-antihuman, GeneTex Inc. Biozol, Eching, Germany	1:30	donkey-anti-mouse cy-3, Invitrogen	1:200
VEGF	mouse-antihuman, R&D Systems	1:30	donkey-anti-mouse cy-3, Invitrogen	1:200
Vimentin	mouse-antihuman, Dako Cytomation, Hamburg, Germany	1:50	donkey-anti-mouse cy-3, Invitrogen	1:200
Vinculin	mouse-antihuman 1:50, Sigma-Aldrich	1:50	donkey-anti-mouse cy-3, Invitrogen	1:200

**Table 2 ijms-20-01972-t002:** The primer sequences used for RTD-PCR.

Gene Symbol	Species	Gene Name	Amplicon Length (bp **)	Assay ID ***
COLA1	homo sapiens	type I collagen	66	Hs00164004_m1
DCN	homo sapiens	decorin	77	Hs00370384_m1
PRG4	homo sapiens	lubricin	65	Hs00195140_m1
SCXB	homo sapiens	scleraxis homolog B	63	Hs03054634_g1
TNC	homo sapiens	tenascin C	61	Hs01115665_m1
HPRT	homo sapiens	hypoxanthine-guanine phosphoribosyltransferase	100	Hs99999909_m1

** Base pairs, *** All primers were obtained from Applied Biosystems.

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
