# Peer review of "Migrating Myofibroblastic Iliotibial Band-Derived Fibroblasts Represent a Promising Cell Source for Ligament Reconstruction"

_ijms, 2019, doi:10.3390/ijms20081972_

Reviewer 1 Report

Even though the authors addressed several of my concerns, I still think the manuscript needs to improved before publication. Still there is no quantitative data available in the manuscript regarding the expression pattern of the described cells.

As the authors now omit the immunofluorescence with the Scleaxis antibody, no tendon marker (besides Tenascin C, which is not tendon specific) is shown. As the main merit of this paper would be to add to our knowledge on tendon cell properties and characterization, this is a major limitation.

Regarding supplemental figure 1, I have massive concerns about the statistical interpretation of the data. It seems statistical significance is only reached due to including one obvious outlier. I am aware sampling human material from young donors is challenging, and any data obtained is valuable. However, in this case it seems over-interpreted.

Author Response

1. We added quantitative RTD-PCR data for scleraxis, tenascin C, type I collagen, decorin and lubricin now (supporting the removed scleraxis immunostaining) as supplemental figure 2. Respective method description (4.8) and primer sequences (novel table 2), description of the results and discussion of data are added.

2. We added quantitative RTD-PCR data for scleraxis now as supplemental figure 2.

3. I am reluctant to designate the sample from the very young individual as an outlier – I think it makes the donor collective unique since there exist no data from such a young ITB donor. Therefore, this limitation that the significant interrelation between age and cellularity seems to depend on the inclusion of the 5 weeks old baby is stated in the manuscript:

“However, the interrelation between age and cellularity was not any longer significant when the immature ITB sample from the very young children was excluded” (see 2.1). This remark was also added in the respective figure legend (suppl. Figure 1): However, in response to exclusion of the very young donor (5 weeks) the interrelation was not any longer significant.

And also the discussion section is adapted accordingly: Interestingly, the observation of a lower cellularity in samples from older compared to younger donors as reported by others [41, 42] could only confirmed here if the very young donor was included in analyses.

Reviewer 2 Report

The authors have addressed the majority of comments satisfactorily and the manuscript is improved. I have a couple of additional comments on the revised manuscript:

Figure 5: The figure legend states that some of the explants were derived from rabbit but this information is not included in the methods - please clarify. Please also correct labels on figure.

Methods: It is not clear from the methods how many donors were used for each experiment. Please include n numbers for separate experiments.

Author Response

1. We added this donor in the method part (4.1): „ACL explants derived from adult female rabbit (New Zealand rabbits, knees obtained from the abbatoir) were also included into histological analysis.” We translated the labelling of the figure.

2. For all methods at least three independent experiments with cells from three donors were performed. Histopathology: the replicates have already been detailed before. For the other methods it was added now.

Round  2

Reviewer 1 Report

The authors have addressed all my concerns, I now recommend the manuscript for publikation.